# Agronomic Behavior of Mexican Roselle Cultivars Produced under Protected Agricultural Conditions

**DOI:** 10.3390/plants11202767

**Published:** 2022-10-19

**Authors:** Jeny Hinojosa-Gómez, César San Martín-Hernández, Tomás Osuna-Enciso, José B. Heredia, Josefina León-Félix, María D. Muy-Rangel

**Affiliations:** 1Centro de Investigación en Alimentación y Desarrollo, A.C. (CIAD), Coordinación Culiacán, Carretera a Eldorado Km 5.5 Campo El Diez, Culiacán 80110, Mexico; 2Colegio de Postgraduados, Carretera México-Texcoco km 36.5. Montecillo, C.P., Texcoco 56230, Mexico

**Keywords:** growth dynamics, *Hibiscus sabdariffa*, hydroponics, minerals, morphological characteristics

## Abstract

Growth dynamics and morphological traits are used to define the characteristics of roselle cultivars *Hibiscus sabdariffa*. The morpho–agronomic variability of plants was evaluated, as well as the concentration of macro and micronutrients in leaves of Mexican roselle cultivars in hydroponic and greenhouse conditions. The following roselle cultivars were studied: ‘Cruza Negra’, ‘UAN16-2′, ‘Criolla Huajicori’, ‘UAN 6 Puga’, ‘UAN 25-1′ and ‘4Q4′. The relative growth rate of the crops was fitted to a positive cubic, negative quadratic and positive linear model, whereas stem diameter, fitted to a linear model, had a negative quadratic response. The foliar surface of the cultivars ‘4Q4’, ‘Cruza Negra’, ‘UAN16-2’ and ‘Criolla Huajicori’ was directly related to the size of the flower, the calyx and the corolla. Nitrogen and potassium are the elements that showed the highest concentration in the cultivars Criolla Huajicori and Cruza Negra, while in the other four cultivars the highest concentrations of minerals in the leaf were nitrogen and calcium. The leaves of cultivar 4Q4 provided the greatest amount of minerals, with 83,565 mg kg^−1^ dry basis. Criolla Huajicori was the cultivar that exhibited the tallest height, the most productive branches and the greatest calyx number per plant; however, Cruza Negra exhibited the highest yield by having the longest calyx.

## 1. Introduction

*Hibiscus sabdariffa* L. is an annual plant native to Africa that grows in tropical and subtropical regions of Sudan, Asia and America; known as roselle, it belongs to the Malvaceae family, and has a vegetative cycle of 136 to 168 days after germination depending on the variety [1]. The roselle plant has an erect and semi-woody stem, with leaves of three to five lobes that are toothed and alternating with long, straight petioles. Flowers with five petals are bisexual, solitary, axillary, sessile and yellow or pink. The epicalyx is lanceolate and linear, with 8 to 12 segments, and is attached to a reddish and fleshy calyx composed of 5 segments separated in the form of a cup, which is the object of economic interest for producers. The fruit is an ovoid cap covered by a calyx of 5 lobes that contains an average of 20 seeds [2,3].

The leaves of *H. sabdariffa* are used for the preparation of fresh salads or soups in Nigeria [4], given their high fiber and protein content [5]. In addition, they are rich in oligoelements such as phosphorus, magnesium, calcium, potassium, iron, sodium and zinc [6,7]. Minerals play an important role in human nutrition [8]. Calcium and magnesium are involved in metabolic processes, and phosphorus is necessary for biochemical reactions in which phosphates such as ATP serve as substrates or products [9,10,11]. Therefore, it is important to promote the consumption of minerals, and vegetables are a primary source. In addition, in terms of bioactivity, extracts of roselle leaves have been shown to exhibit both anticarcinogenic and antilipidaemic effects [12,13] attributed to hibiscus acid, sitosterol- β-D-galactoside, pectin, quercetin, kaempferol and other flavonoids [14]. The calyx of the roselle fruit is the structure of the plant that is most marketed, as it has an excellent flavour and aroma and is an important source of pigments, minerals and dietary fiber [15,16,17]. Additionally, it is reported to have a high content of phenolic compounds and flavonoids, of which anthocyanins are prominent, conferring nutraceutical and pharmacological properties and serving as natural pigments [18,19,20,21]. In addition to the sensory properties of roselle, the consumption of its chemical compounds containing beneficial properties increases its demand in terms of enhancing human health and wellbeing and becoming increasingly economically important for many countries such as Sudan, Mexico, China, Taiwan, Thailand and India [22,23], thus intensifying its production in greenhouses [24]. However, both the morphological and biochemical characteristics of the crop can vary by phenotype or genotype and even more so as a result of the growing conditions.

The generalized method for defining the variability of roselle cultivars is the study of morpho–agronomic traits because the morphological characteristics determine its classification [25]. Direct measurements of growth, height, leaf area and cultivation time are used for the quantitative analysis of plant growth and for the calculation of crop growth rates of commercial importance [26].

Multiple studies have reported on the chemical composition of roselle [5,17,20,21]. However, there is a need to investigate the growth of different cultivars to propose models that can predict the behaviour of growth rates in the vegetative and reproductive stages, as well as the related morphological variables. This study evaluated the morpho–agronomic variability of the plants as well as the composition of macro and micronutrients in the leaves of six cultivars of Mexican roselle grown in hydroponic and greenhouse conditions. The hypothesis of this research work is that the growth dynamics of roselle, the morpho–agronomic characteristics and the composition of macro and micronutrients in the leaves of six cultivars of Mexican roselle is dependent on the type of cultivar and not on the agronomic conditions of the crop.

## 2. Results and Discussion

### 2.1. Growth Rates

The behaviour of the relative crop growth rate (RGRC) for the six roselle cultivars was fitted to the third-order polynomial regression model, resulting in a positive cubic, negative quadratic and positive linear model for the effect of days after transplantation (DAT) (Table 1).

The roselle variety Criolla Huajicori developed the highest RGRC with 3.25 cm day^−1^ at 45 DAT, whereas UAN 16-2, UAN 25-1 and Cruza Negra reached 3.16, 2.89 and 2.78 cm day^−1^, respectively, at 36 DAT; UAN 6 Puga exhibited the lowest growth (Figure 1).

The growth rate of plants decreased due to the formation and development of reproductive structures from the time of flowering as these organs became the primary physiological demand on the plant, accelerating its development as a deciduous plant [27,28]. Vegetative growth was between 91 and 106 DAT for the six roselle cultivars (Figure 1).

The high RGRC was related to a taller height (245.4 cm) (Table 2), a higher calyx number and a greater number of productive branches, all desirable characteristics in the cultivar Criolla Huajicori.

Several antecedents have exhibited variability in the height of roselle plants, with values of 30 to 192 cm under open-field cultivation conditions [29,30,31,32], and between 83 and 114 cm in a greenhouse cultivation setting [33]. The cultivars of roselle in this study have similar or greater heights than those reported previously (Table 2).

The relative growth rate of the stem diameter (RGRSD) of roselle plants fits a linear increasing model and exhibits a negative quadratic response to DAT (Table 1), which describes the growth pattern up to 78 DAT (Figure 2).

The maximum RGRST was presented by cultivar 4Q4 with 0.24 mm day^−1^, reaching a diameter of 17 mm (Table 2) and a rate of up to 37% higher than UAN 6 Puga, the cultivar that exhibited the lowest rate of all evaluated samples. All cultivars ceased stem growth at 76 DAT, the period in which flowering ends and plant fruiting begins (Figure 2). Stems are the most important part of the plant because they help control their growth by functioning as nutrient conductors and serving as support. Stems of greater thickness are necessary for the support of branches and fruits, as well as being indicators of good productivity [34].

In other roselle cultivars, thick stems of 46 mm in diameter have been found [35], although, on average, all roselle plant cultivars have stem diameters of 17 to 28 mm [30,36]. Of the roselle samples from our study, only 4Q4 is similar to the aforementioned values, and the remainder developed smaller stem diameters (Table 2).

### 2.2. Growth Variables

Criolla Huajicori was the cultivar that exhibited the greatest number of productive branches per plant, with 10 branches more than the smallest number, as observed in UAN 6-Puga (Table 2), which is related to a greater plant height. Cultivars in open fields exhibited values of between 7 and 13 productive branches for cultivars from Iraq and Havana Kadhem Abbas [32,36], whereas for the roselle cultivar in greenhouse conditions, between 5 and 12 productive branches per plant have been found [33]. All these values are within those reported in the present research. It should be noted that the largest number of branches generated more flowers per plant, as observed in the cultivar Criolla Huajicori (Table 2 and Table 3).

### 2.3. Roselle Flower Measurements

Figure 3 shows that the external diameter of the corolla of roselle flowers varied between 49 and 64 mm, with cultivar 4Q4 having the largest statistically significant size and cultivar Huajicori having the smallest diameter, which is not related to the length of the flower. In addition, it is important to mention that the remaining cultivars exhibited no significant differences. The internal corolla diameter exhibited significant differences with maximum measurements of 21.1 mm for cultivar 4Q4. The flower length presented values between 42.6 and 51.3 mm for the cultivars Criolla Huajicori and 4Q4, respectively. Larger calyx length values, in the presence of a flower, were observed in Cruza Negra and UAN 6-Puga (34.1 and 32.8 mm, respectively), and the length of the epicalyx ranged between 16 and 25.9 mm, the highest value of which was observed in Cruza Negra (Figure 3).

### 2.4. Roselle Calyx and Capsule Measurements

Statistically, cultivars UAN 25-1 and 4Q4 had the largest calyx diameters; in contrast, the smallest were recorded in Criolla Huajicori, which also had the shortest calyx length (Figure 4). The results of this study are within the values of diameters (20.9 and 57.8 mm) and calyx length (34 and 87 mm) reported for different roselle cultivars [29,35,37]. It would be expected that the roselle samples with a larger capsule diameter would be those with a larger calyx diameter, but the greatest length was achieved by Cruza Negra and UAN 16-2 (Figure 4).

Capsule length was 22.7 to 26.6 mm, lower than the previously reported values, but greater than the diameter (14.38 mm) of 13 genotypes of roselle from Guatemala [38]. The number of seeds per capsule oscillated between 16 and 20 and was independent of capsule size (Figure 4). However, there are reports of 33 seeds per capsule for cultivars from Guatemala, Nigeria, Malaysia and India, which is economically important for the reproduction of the material [30,38,39].

### 2.5. Leaf Size and Leaf Surface Area

Figure 5 presents a significant difference in the petiole length of roselle leaves (between 83.8 and 99.5 mm), where the greatest length was observed in 4Q4, the opposite being true for UAN 6-Puga. Values between 90 and 115 mm for the leaf petiole of 13 Guatemalan genotypes have been reported [38]. The length and width of the central lobe of the roselle leaves was greatest for the cultivar Cruza Negra (Figure 5).

Similar leaf lengths were reported for cultivars from Guatemala [38]. The average length of the lateral lobe in the leaves was highest for Cruza Negra (114.8 mm) and lowest for UAN 6-Puga (85.7 mm); the width of the lateral lobe was greatest for 4Q4 (35.6 mm), being statistically different from the rest of the cultivars (Figure 5). Measurements of the central lobe width of 47 cultivars in Guerrero, the average of which was 17.8 mm, resulted in values that were up to 70% less than those of the present study [35].

The leaf surface area values were between 76.4 and 91.3 cm^2^, with UAN 16-2, Cruza Negra, 4Q4 and Criolla Huajicori being the cultivars with the largest leaf surface area (Table 2). Leaves with larger leaf surface areas have a greater capacity to acquire solar energy and carbon dioxide, and therefore obtain a greater carbon gain [40]. Thus, a high RGRC was observed in UAN 16-2, 4Q4 and Criolla Huajicori (Figure 1). Accordingly, several studies with woody species have identified a positive correlation between RGRC and leaf ratio [27,40]. Regarding this characteristic, an Indian cultivar had larger leaves, with an average leaf surface area between 93.1 and 101.4 cm^2^ [29]. However, the evaluated morphological characteristics and yield were lower than those reported in the present study.

### 2.6. Yield of Roselle Cultivars

The number of calyces per plant oscillated between 49 and 173, corresponding to UAN 6-Puga and Criolla Huajicori, respectively (Table 3). Accordingly, various studies have indicated that this characteristic depends on the cultivar type and the location where the plant is developed, represented by the low (9 and 24), intermediate (18 to 54) and high (89 to 441) numbers of calyces for roselles grown in Egypt–India [24,41], Iraq [31,32] and Guatemala–Brazil [38,42], respectively.

Statistically, roselle cultivars Cruza Negra, Criolla Huajicori and UAN 25-1 developed a similar number of calyces. However, Cruza Negra statistically presented a heavier and higher yield of calyces on a fresh and dry weight basis per ha (Table 3). This indicates that these variables are dependent on the size of the calyces and the total solids that they develop during their growth (Figure 3). With the exception of the cultivar UAN-6 Puga, all the roselles of the present study developed dry weight values of 910 to 1,872 kg ha^−1^, which is within the range (480 and 2522 kg ha^−1^) that is reported for cultivars from Brazil and Nigeria [29,42].

### 2.7. Number of Days in the Roselle Development Cycle

The time between the germination period and flowering of the roselle plants was between 42 and 51 days, the cultivars from the present study being the fastest to develop compared to the cultivars reported in open-field conditions in Brazil (100 days), India (115 days) and Guatemala (120 to 141 days) [38,41,42]. The earliest cultivar was Cruza Negra at 148 days from germination to harvest, which was significantly different from the later cultivars UAN 25-1 and UAN 6-Puga (Table 4), being the most precocious in comparison to roselles in open-field conditions, measuring 149 to 203 days to harvest, respectively [38,41,42]. This represents an advantage for producers because maximum yields may be obtained from shorter development times, in addition to reducing crop maintenance costs.

### 2.8. Mineral Composition of Roselle Leaves

The mineral composition in roselle leaves varies by cultivar (Table 5). Based on mineral content and assuming a consumed amount of 25 g of dehydrated roselle leaves, the minerals P, Ca, Mg, Mn and Zn contributed most to nutritional intake based on the recommended daily intake (RDI) [43]. Among the most significant minerals for all roselle cultivars were Mn and P, reaching values greater than 260 mg day^−1^ and 40% of the RDI. The Ca content was a high as 51% of the RDI for cultivar 4Q4. Cruza Negra and UAN 25-1 contained 47% of the RDI of Mg, with the latter also providing the maximum RDI level of Fe (25%) among all cultivars, whereas Zn was highest in UAN 6-Puga (90% of the RDI). The Cu content was highest in the cultivar UAN 16-2, with 23% of the RDI.

The greatest capacity to absorb and assimilate nitrogen in roselle leaves was achieved by the cultivar Criolla Huajicori, being 19% higher than Cruza Negra, which had the lowest content (Table 5). The mineral content of the leaves of the studied cultivars was similar or greater than that reported [24] for an Egyptian cultivar under greenhouse conditions with N, P, Fe, Mn and Zn contents of 29,000, 3900, 162, 68 and 51 mg kg^−1^, respectively. Of these, the Cu content was 88% higher compared to the maximum value found in the present study.

The mineral content of roselle leaves produced in open-field conditions is variable with respect to the samples studied. Cultivars from Nigeria had lower contents of Fe, Mg, Zn, Na and K of 18.5 to 34.5, 18.4 to 21.6, 0.03, 2.5 to 6.1 and 38.2 to 61.8 mg kg^−1^, respectively [44,45]. Similar mineral content values for three Nigerian cultivars, Mg (4200 to 4400 mg kg^−1^), Ca (15,000 to 22,000 mg kg^−1^) and K (12,000 to 15,000 mg kg^−1^), have been reported (Atta et al., 2010a), whereas Atta et al. (2010) observed higher values of Fe (180 to 230 mg kg^−1^), Mn (450 to 560 mg kg^−1^) and Cu (45 to 100 mg kg^−1^).

The total mineral content in leaves was 25% higher than that reported [46] in the calyx for the same roselle cultivars and under the same growing conditions, where Mn, Zn, N and Mg exhibited the highest increase (72, 54, 47 and 33%, respectively). Given their mineral content, the consumption of roselle leaves is recommended.

### 2.9. Pearson’s Correlation

In all cultivars, the length of the central lobe, the calyxes harvested per plant and the calyx yield in fresh weight and dry weight showed a positive and highly significant correlation (Table 6). This relationship is due to the fact that K is an element that, according to the phenological stage of a crop, moves to the areas where it is required [47]; the highest content is found in the leaves when the plant is in the process of fruit maturation [48]. In addition, K participates directly in fruit quality, since the content of this element is concentrated between 70 and 90% in the calyxes, this intensive accumulation is mainly due to the phloem sap together with the assimilates [49], resulting in fruit with greater firmness, better organoleptic quality and a larger calibre, for which it correlates directly with harvest yields [50].

Magnesium is one of the elements with the highest mobility in plants and is required for the synthesis of chlorophyll, translocated via the phloem to points of active growth such as flowers and developing leaves [51,52], this explains the positive correlation with the lateral lobe of roselle leaves (Table 6).

Nitrogen actively participates in the development of flowers and fruitful buds, it also brings with it an increase in the photosynthetic rate, increasing the growth of crops since all the dry matter produced by the plant depends totally on this process [53], this is due to the source–sink relationship, where the nitrogen content moves towards the flowers and fruits once the plant’s reproduction stage has begun, with N being the second element most exported towards the fruit after K [54], which explains the high correlation of this element with the length of flowers and calyxes of roselle cultivars (Table 6).

## 3. Materials and Methods

### 3.1. Materials and General Conditions of Cultivation

The experiment was conducted in a chapel type greenhouse from October 2015 to March 2016 (28 m altitude, 24°44′02″ N and 107°27′16″ W). Six cultivars of *H. sabdariffa*: ‘Cruza Negra’ (‘Cruza N ’), ‘UAN16-2’, ‘Criolla Huajicori’ (‘Criolla H.’), ‘UAN 6 Puga’, ‘UAN 25-1’ and ‘4Q4’) were studied, as provided by the Autonomous University of Nayarit, México. All cultivars were grown under hydroponic conditions. Within the greenhouse, a Hobo data logger was placed over the canopy of the plants to record daily temperatures and relative humidity (average maximum and minimum temperature of 23.95 and 20.18 °C, respectively, and relative humidity of 71.49%). The seeds were planted in 120-well germination trays in a peat moss–agrolite mixture (70:30 *v/v*). At 14 days after germination, the seedlings were transplanted to a greenhouse in a completely random arrangement, and each experimental unit represented four plants and four experimental replicates for each cultivar. Each seedling was placed in 13 L polyethylene bags of substrate (60% fluvisol soil and 40% coconut fiber). The distance between bags was 0.50 m and between rows was 1.0 m. A Steiner solution [55] with an EC of 2 dS m^−1^ and pH of 5.8 was provided by drip irrigation at a rate of eight daily events of 5 min^−1^ using 4 L h^−1^ drippers.

### 3.2. Growth Rates

Growth rates (cm, mm day^−1^) were evaluated every seven days during the vegetative and reproductive stages using the relative crop growth rate (RGRC) and stem diameter relative growth rate (RGRSD), according to [56], as follows:(1)$$\mathrm{RGRC}=\left(\frac{{\mathrm{HS}}_{2}-{\mathrm{HS}}_{1}}{t_{2}-t_{1}}\right)$$
(2)$$\mathrm{RGRSD}=\left(\frac{{\mathrm{DS}}_{2}-{\mathrm{DS}}_{1}}{t_{2}-t_{1}}\right)$$
where HS_1_ and HS_2_ are the initial and final stem heights, respectively; DS_1_ and DS_2_ are the initial and final stem diameters of the time interval, respectively; and t_1_ and t_2_ are the initial and final times, respectively.

### 3.3. Morphological Parameters

When the plants reached at least 50% anthesis, 10 flowers were selected per plant to determine the internal and external diameter of the corolla, as well as the length of the flower, calyx and epicalyx. From the leaves used to estimate the leaf surface area, the length and thickness of the petiole, as well as the length and width of the lateral and central lobes, were measured. Additionally, the number of productive branches, and the heights of the first flower and first branch were evaluated. The harvesting of the calyces was performed in the commercial maturity stage (the opening of the capsule, which makes the seeds visible). The entire plant was cut, and the calyces of the capsules were manually separated. The number of calyces per plant, the diameter and the length of the calyx and capsules, the number of seeds and the weight of fresh and dried calyces were analysed, expressing the yield in grams per plant and kilograms per hectare, both on a wet and dry basis. In addition, the time (days) for anthesis and harvest of the roselle calyces were evaluated from germination/emergence.

### 3.4. Leaf Surface Area

To estimate leaf surface area, 12 fully expanded leaves were collected halfway up the height of the plant and from the productive branches for each cultivar; the leaf surface area (cm^2^) was calculated non-destructively using a photocopier [57].

### 3.5. Leaf Analysis

The macro and microelement contents of the leaves previously collected were evaluated. The total N was evaluated using the micro-Kjeldahl method [58], and phosphorus was evaluated by visible spectrophotometry (spectrophotometer UV/Vis 6705 Jenway^®^, St Neots, UK). The micronutrients Ca, Mg, K, Na, Fe, Zn, Mn and Cu were quantified by atomic absorption spectrometry (AA FS flame AA 280FS + SIPS 20, Agilent Technologies^®^, Santa Clara, CA, USA) according to official methods [59].

### 3.6. Statistical Analysis

The leaf surface area, the morphological parameters and the content of macro and micronutrients were analysed using analysis of variance and Tukey’s comparison of means test (*p* ≤ 0.05). The initial regression models were specified for growth rates based on the responses of the variables of interest (RGRC and RGRSD). Sign directions and interactions were considered until a best fit model was obtained, using the mean squared error (MSE) as the criterion for goodness-of-fit [60]. Pearson correlations between mineral content and morphological characteristics of roselle cultivars were calculated, and correlations with values greater than 0.8 and highly significant (*p* ≤ 0.001) were selected. The software program MINITAB 2014 version 17.0 was used to perform the data analysis.

## 4. Conclusions

Differences were found in the behaviour of the growth dynamics of all roselle cultivars under greenhouse conditions. The first study on the relative growth rate of the roselle cultivar that fits a positive cubic, negative quadratic and positive linear model is reported, whereas the growth rate of the stem diameter fits a negative linear and quadratic model, in terms of the effects of days after transplant. Produced hydroponically, the cultivar Cruz Negra may be used for commercial production due to its high calyx yield, and the consumption of roselle leaves is recommended due to their high mineral content, conferring nutritional benefits for human health.

This research shows that the morpho–agronomic selection of roselle cultivars under hydroponic conditions allows the selection of new cultivars for commercial exploitation with characteristics desired by roselle producers, in addition to defining the phenological stages according to growth rates, which allows for improving the agronomic management of the crop during all its development stages. It is important to continue with this type of research, especially in new roselle crosses, in order to define their possible agronomic and commercial potential before being used commercially.

## Figures and Tables

**Figure 1 plants-11-02767-f001:**
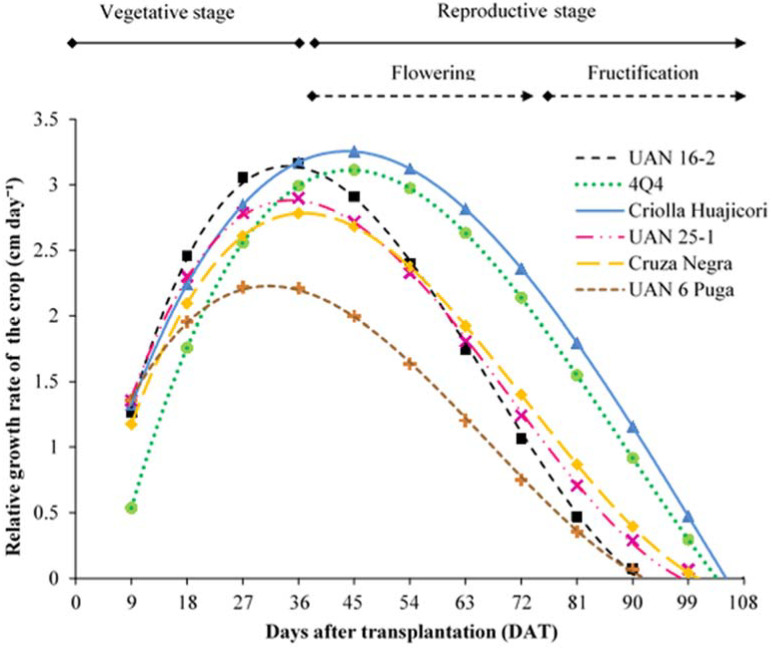
Relative growth rate of the crop (RGRC) of roselle cultivars as a function of time. Estimated values from models (Table 1).

**Figure 2 plants-11-02767-f002:**
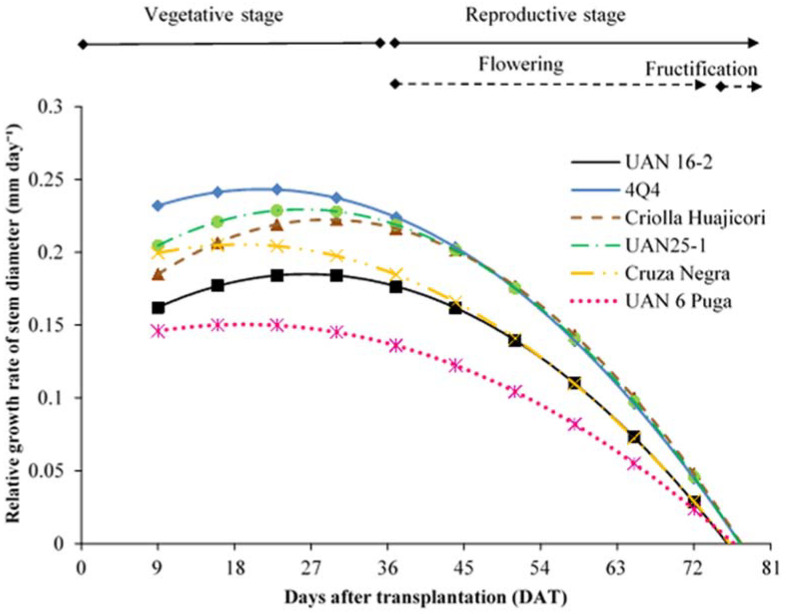
Relative growth rate of stem diameter of roselle cultivars, RGRSD, as a function of time. Estimated values from models (Table 1).

**Figure 3 plants-11-02767-f003:**
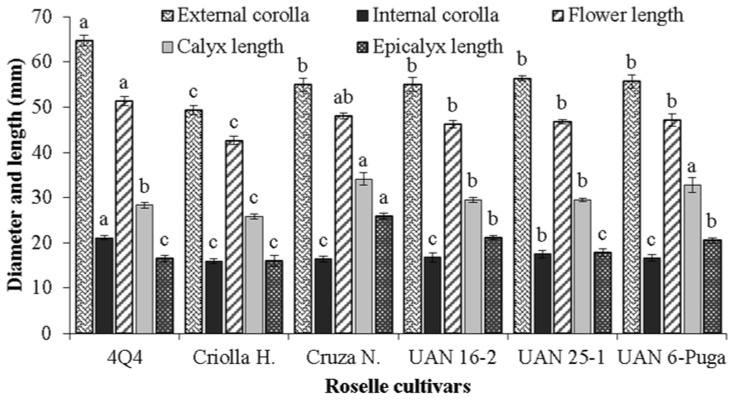
External and internal diameter of corolla and length of flower, calyx and epicalyx of different cultivars of roselle. Different letters on bars corresponding to the same response variable indicate significant differences between cultivars according to the Tukey test (*p* ≤ 0.05). Vertical bars correspond to the standard deviation.

**Figure 4 plants-11-02767-f004:**
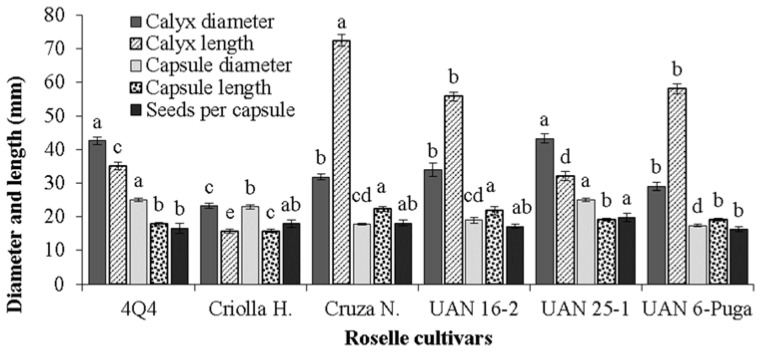
Diameter and length of calyx and capsule and number of seeds per capsule of different roselle cultivars. Different letters on bars corresponding to the same response variable indicate significant differences between cultivars according to the Tukey test (*p* ≤ 0.05). Vertical bars correspond to the standard deviation.

**Figure 5 plants-11-02767-f005:**
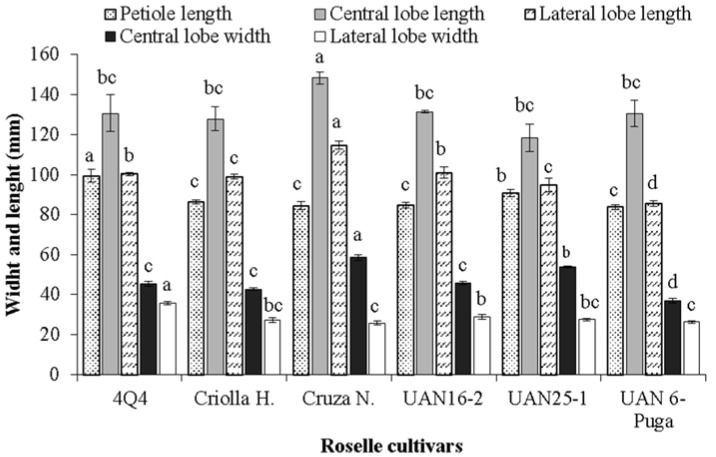
Length and width of lobes and foliar surface area of different cultivars of roselle. Different letters on bars corresponding to the same response variable indicate significant differences between cultivars according to the Tukey test (*p* ≤ 0.05). Vertical bars correspond to the standard deviation.

**Table 1 plants-11-02767-t001:** Best-fit models of growth rates for the roselle cultivars cultivated hydroponically in a greenhouse.

Growth Rate	Cultivar	Model		*p*-Value
RGRC *	UAN 16-2	Y = −0.64686 + 0.2557DAT − 0.0051 DAT^2^ + 0.0000260DAT^3^		<0.0001
		MSE ^a^ = 0.22750	R^2^ = 0.8573	
	4Q4	Y = −1.16136 + 0.21702DAT − 0.00327DAT^2^ + 0.00001239DAT^3^		<0.0001
		MSE = 0.12680	R^2^ = 0.9148	
	Criolla H.	Y = 0.05443 + 0.16121DAT − 0.00235DAT^2^ + 0.00000772DAT^3^		<0.0001
		MSE = 0.23711	R^2^ = 0.98435	
	UAN 25-1	Y = −0.12310 + 0.19722DAT − 0.00382DAT^2^ + 0.00001866DAT^3^		<0.0001
		MSE = 0.25238	R^2^ = 0.8141	
	Cruza N.	Y = −0.21955 + 0.18356DAT − 0.00333DAT^2^ + 0.00001518DAT^3^		<0.0001
		MSE = 0.09309	R^2^ = 0.9186	
	UAN 6 PUGA	Y = 0.35909 + 0.13533DAT − 0.00286DAT^2^ + 0.00001467DAT^3^		<0.0001
		MSE = 0.12374	R^2^ = 0.8537	
RGRSD *	UAN 16-2	Y = 0.13227 + 0.00399DAT − 0.00007546DAT^2^		<0.0001
		MSE = 0.00166	R^2^ = 0.7774	
	4Q4	Y = 0.20901 + 0.00325DAT − 0.00007679DAT^2^		<0.0001
		MSE = 0.00175	R^2^ = 0.8313	
	Criolla H.	Y = 0.14337 + 0.00546DAT − 0.00009423DAT^2^		<0.0001
		MSE = 0.00197	R^2^ = 0.7722	
	UAN 25-1	Y = 0.17113 + 0.00450DAT − 0.00008664DAT^2^		<0.0001
		MSE = 0.00096	R^2^ = 0.8884	
	Cruza N.	Y = 0.18355 + 0.00233DAT − 0.00006217DAT^2^		<0.0001
		MSE = 0.00134	R^2^ = 0.8419	
	UAN 6 PUGA	Y = 0.13392 + 0.00173DAT − 0.00004531DAT^2^		<0.0001
		MSE = 0.00102	R^2^ = 0.7764	

^a^ Mean square error. * RGRC = Relative crop growth rate, RGRSD = Stem diameter relative growth rate.

**Table 2 plants-11-02767-t002:** Number of productive branches, height, stem diameter and foliar surface area of different roselle cultivars produced in greenhouse hydroponics.

Cultivar	Num. of Productive Branches	Height (cm)	Stem Diameter (mm)	Foliar Surface Area (cm^2^)
4Q4	8.1 ± 0.5 ^a^ b *	200.7 ± 13.7 b	17.2 ± 1.0 a	89.9 ± 1.3 a
Criolla H.	14.9 ± 1.2 a	245.4 ± 22.4 a	15.9 ± 1.7 ab	86.9 ± 1.6 c
Cruza N.	5.9 ± 0.5 c	176.5 ± 10.8 c	14.3 ± 1.9 bc	88 ± 1.5 a
UAN 16-2	7.7 ± 0.9 b	173.2 ± 23.8 c	13.5 ± 1.6 c	91.3 ± 0.9 a
UAN 25-1	8.2 ± 0.7 b	173.8 ± 15.1 c	15.3 ± 1.2 b	77.1 ± 1.9 b
UAN 6-Puga	4.8 ± 0.3 c	129.8 ± 17.1 d	11.5 ± 1.1 d	76.4 ± 2.2 b

^a^ Standard deviation. * Values are mean ± SD (n = 16) different letters in the same column indicate significant differences between samples using the Tukey test (*p* ≤ 0.05).

**Table 3 plants-11-02767-t003:** Yield of different roselle cultivars produced in greenhouse hydroponics.

Cultivar	Calyces Harvested per Plant	Fresh Weight Calyx	Fresh Weight Yield Calyx	Dry Weight Calyx Yield
		**kg ha^−1^**		
4Q4	96.8 ± 21.1 ^a^ bc *	0.63 ± 0.04 c	5551 ± 378 c	971.62 ± 80.7 b
Criolla H.	173.5 ± 39.3 a	0.91 ± 0.05 b	7900 ± 410 b	989.67 ± 50.9 b
Cruza N.	116.4 ± 39.8 ab	1.47 ± 0.11 a	12,855 ± 977 a	1872.43 ± 119.3 a
UAN 16-2	90.4 ± 29.5 bc	0.77 ± 0.04 bc	6720 ± 343 bc	910.51 ± 29.8 b
UAN 25-1	120.3 ± 23.9 ab	0.88 ± 0.07 b	7713 ± 614 b	946.19 ± 72.8 b
UAN 6-Puga	49.0 ± 12.13 c	0.24 ± 0.02 d	2077.9 ± 137.7 d	263.9 ± 27.4 c

^a^ Standard deviation. * Values are mean ± SD (n = 3) different letters in the same row indicate significant differences between samples using the Tukey test (*p* ≤ 0.05).

**Table 4 plants-11-02767-t004:** Development of different cultivars of roselle produced in greenhouse hydroponics.

Cultivar	Days to Flowering	Days to Harvest
4Q4	48.7 ± 1.3 ^a^ ab *	154.1 ± 1.3 bc
Criolla H.	51.2 ± 2.1 a	156.4 ± 0.4 ab
Cruza N.	46.7 ± 1.5 bc	148.3 ± 2.2 d
UAN 16-2	43.1 ± 1.1 d	151.9 ± 0.5 c
UAN 25-1	44.9 ± 1.1 cd	156.9 ± 0.6 a
UAN 6-Puga	42.3 ± 0.7 d	157.9 ± 1.1 a

^a^ Standard deviation. * Values are mean ± SD (n = 16) different letters in the same column indicate significant differences between samples using the Tukey test (*p* ≤ 0.05).

**Table 5 plants-11-02767-t005:** Mineral content (dry basis) of leaves from different roselle cultivars.

	4Q4	Criolla H.	Cruza N.	UAN 16-2	UAN 25-1	UAN 6-Puga	RDI ^a^
	**------------------------------------------------- mg kg^−1^ ---------------------------------------------------**						**mg d^−1^**
N	36,594 ± 262 ^b^ bc *	38,502 ± 694 a	31,158 ± 874 d	36,856 ± 312 b	36,808 ± 652 b	35,320 ± 582 c	-
P	10,663 ± 67.9 c	11,747 ± 43.3 b	11,929 ± 91.5 a	11,832 ± 64.7 ab	11,702 ± 32.6 b	10,663 ± 18 c	664
K	13,397 ± 104 b	13,259 ± 233 b	18,171 ± 546 a	11,365 ± 542 c	10,848 ± 94 c	7554 ± 302 d	4700
Ca	18,364 ± 267 a	10,648 ± 255 d	10,527 ± 346 d	13,291 ± 90 c	12,716 ± 227 c	14,346 ± 373 b	900
Mg	3813 ± 103.8 c	3819 ± 65.9 c	4722 ± 110.8 a	4107 ± 79 b	4739 ± 183 a	3820 ± 101 c	248
Na	76.7 ± 2.3 c	86.0 ± 3.0 b	88.3 ± 1.2 b	97.6 ± 2.0 a	75.6 ± 2.0 c	52.6 ± 3.9 d	1500
Fe	130.8 ± 9.3 bc	108.2 ± 4.3 c	154.1 ± 14.2 ab	147.0 ± 17.2 b	175.5 ± 8.4 a	150.4 ± 11.7 ab	17
Mn	309.4 ± 11.8 bc	289.1 ± 11.4 c	355.0 ± 17.6 ab	283.6 ± 11.5 c	323.1 ± 16.1 abc	364.6 ± 39.7 a	2
Zn	207.7 ± 8.2 c	289.6 ± 10.8 b	205.8 ± 10.8 c	300.9 ± 16.1 b	314.3 ± 8.6 b	363.3 ± 11.3 a	10
Cu	8.3 ± 0.6 a	8.3 ± 1.0 a	5.5 ± 0.3 b	9.3 ± 0.8 a	8.2 ± 1.3 a	8.3 ± 0.3 a	1
Total	83,565.6	78,756.3	77,316.2	78,290.8	77,711.4	72,643.4	

^a^ Recommended daily intake. ^b^ Standard deviation. * Values are mean ± SD (n = 3) different letters in the same row indicate significant differences between samples using the Tukey test (*p* ≤ 0.05).

**Table 6 plants-11-02767-t006:** Main Pearson correlation coefficients between mineral content and morphological characteristics of roselle cultivars.

	Flower Length (mm)	Calyx Length (mm)	Central Lobe Length (mm)	Lateral Lobe Length (mm)	Calyces Harvested per Plant	Fresh Weight Yield Calyx (kg ha^−1^)	Dry Weight Calyx Yield (kg ha^−1^)
K	0.121	0.407	0.942 *	0.483	0.894 *	0.894 *	0.96 *
Mg	0.456	0.522	0.459	0.842 *	0.676	0.676	0.606
N	0.836 *	0.864 *	0.53	0.653	0.501	0.501	0.579

* = highly significant (*p* ≤ 0.001).

## Data Availability

Not applicable.

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
