# Peer review of "Agronomic Behavior of Mexican Roselle Cultivars Produced under Protected Agricultural Conditions"

_plants, 2022, doi:10.3390/plants11202767_

Round 1

Reviewer 2 Report

The work of Hinojosa-Gómez et al. “Agronomic behaviour of mexican roselle cultivars produced under protected agricultural condition” presents an interesting analysis of the morpho-agronomic variability of cultivars under hydroponic condition. Mineral analysis of the leaves is also discussed. The paper is well organized and carefully written. The discussion of the data is based on recent bibliography. It presents relevant and worthy findings about this crop in commercial greenhouses. Moreover, the study is interesting and of practical value as there are very few studies about Mexican roselle cultivars.

As suggestions to improve the paper, I include the following commentaries:

Abstract should be improved: remove statistical analysis; abstract should include the main conclusions of the study, please include which minerals are the most important for each cultivar (not only the total minerals).

However, the paper is very descriptive and do not explain the physiological processes behind the morphological or mineral composition responses. This is an interesting paper but, in my opinion, out of the main scope of Plants and more related with the scope of Horticulturae or similar journals.

If the editor thinks it should be published in Plants, then the discussion should be partially rewritten to give it a slightly more physiological approach.

Round 2

Reviewer 1 Report

I accept the introduced amendments, additions and clarifications.
The article can be published.

Reviewer 2 Report

The manuscript has been revised accordingly. In my opinion, the paper can be accepted for publication.